# Dried Rumen Digesta Pellet Can Enhance Nitrogen Utilization in Thai Native, Wagyu-Crossbred Cattle Fed Rice Straw Based Diets

**DOI:** 10.3390/ani10010056

**Published:** 2019-12-26

**Authors:** Anuthida Seankamsorn, Anusorn Cherdthong

**Affiliations:** Tropical Feed Resources Research and Development Center (TROFREC), Department of Animal Science, Faculty of Agriculture, Khon Kaen University, Khon Kaen 40002, Thailand; aontoranu@gmail.com

**Keywords:** rumen content, environmental pollution, rumen microorganism, ruminant

## Abstract

**Simple Summary:**

Generated rumen digesta is wasted and becomes an environmental contaminant in most slaughterhouses in Thailand. Dried rumen digesta (DRD) is a mixture of digestible and indigestible feed residues and are fairly rich in nutrients. DRD has the capacity to become an alternative protein source for ruminants’ diets. DRD in pellet form could be an alternative strategic supplement for Thai-native, Wagyu-crossbred cattle to enhance N balances and microorganisms.

**Abstract:**

The goal of the current research was to study the effects of a diet of dried rumen digesta pellets (DRDP) on diet utilization, ruminal microorganisms, and ruminal microbes in Thai native, Wagyu-crossbred cattle. Four Thai native, Wagyu-crossbred, beef cattle were assigned to a 4 × 4 Latin square design to supplement DRDP levels at 0, 50, 100, and 150 g/d, respectively. Rice straw intake, total intake, and estimated energy intake varied significantly among the different DRDP levels. Nitrogen intake, apparent N absorption, and apparent N retention were significantly enhanced when compared to the 0 g/d DRDP. DRDP supplementation at 150 g/d produced the greatest apparent digestibility of crude protein compared to the group that was not fed DRDP. Supplementation of DRDP did not alter the population of protozoa, whereas the addition of 150 g DRDP significantly increased the fungal zoospore. Supplementation of DRDP at various levels did not change the concentration of volatile fatty acid (VFA) or the VFA profiles. Thus, DRDP could be an alternative strategic supplement for Thai-native, Wagyu-crossbred cattle in order to enhance N utilization and fungal zoospores.

## 1. Introduction

Beef consumption in Thailand has increased over the years with the increasing per capita income. A number of successful enterprises finishing cattle have started to supply high-quality beef [1]. However, the production of beef cattle in Thailand often faces inadequate quantities of nutrients, especially during the dry season, and is characterized by low performance efficiency [2]. Thus, beef cattle producers are becoming increasingly interested in the utilization of locally available feed resources, particularly animal residues [3,4].

Generated rumen digesta is neglected and represents serious natural contaminants in slaughterhouses [4]. Dried rumen digesta (DRD) is a mixture of digestible and indigestible feed, mainly found in the rumen of ruminant and is fairly rich in nutrients [5]. In a previous study, Cherdthong et al. [4] reported that DRD consisted of crude protein at 19.4% dry matter (DM). Moreover, Okpanachi et al. [6] revealed that DRD contained microorganism cells, minerals, and fermentation end products without negative physiological factors, which help ruminal ecosystems and improve the potential for microbial growth in the rumen. Adeniji and Balogun [7] reported that DRD showed the capacity to become an alternative nitrogen source for ruminants’ diets. Cherdthong et al. [4] noted that DRD could be substituted for soybean meal (SBM) as a feed ingredient in enhanced rice straw intake and fiber digestion without affecting the ruminal microorganisms and nitrogen (N) utilization in cattle. Furthermore, feeding swamp buffalo with DRD pellets (DRDP) at 150 g/d is suggested, as doing so improves nutrient digestion and could control environmental waste [5]. However, the effect of DRDP on N utilization in ruminants has not yet been reported by Seankamsorn et al. [5]. It is, theoretically, due to pellet diets providing a synchronous supply of N and energy to the rumen, which can enhance the potential of microorganisms to utilize adenosine triphosphate (ATP) and capture N for microbial synthesis [8,9].

It was hypothesized that DRDP could potentially improve N utilization and ruminal microorganisms in cattle. Therefore, the goal of the current experiment was to determine the effects of a diet of DRDP on N utilization, rumen characteristics, and ruminal microbes in Thai native, Wagyu-crossbred cattle.

## 2. Materials and Methods

Thai native, Wagyu-crossbred cattle used in the current work were approved by the Animal Ethics Committee of Khon Kaen University (KKU), approval no. ACUC-KKU 16/2558.

### 2.1. Experimental Feeds

DRD in the present work was detailed by Cherdthong et al. [4] who, in brief, collected the fresh form of ruminal digesta from a local slaughterhouse in Khon Kaen Province, Thailand. Rumen digesta was obtained from cattle fed on rice straw or native grass based diets with additional small level of concentrate. Rumen digesta was sundried for two days and ground (1 mm screen using the Cyclotech Mill, Tecator, Sweden). Then, the mixed DRD was mixed with other ingredients. Finally, all ingredients were pelleted using a machine. The pellets were sun-dried for two days before being fed to the cattle. The same batch of DRDP was used throughout the experiment. Table 1 presents the ingredients and chemical constituent of the cattle feeds.

### 2.2. Cattle, Treatments, and Feeding Management

Four Thai native, Wagyu-crossbred, beef cattle, each about 1.5-years old, male, with 150 ± 10.0 kg of body weight (BW) were randomly assigned to a 4 × 4 Latin square design to receive DRDP at four levels of 0, 50, 100, and 150 g/d, respectively. The DRDP levels were preformed according to previous work tested in swamp buffaloes [5]. A concentrated diet was provided at 0.5% BW daily with a rice straw based diet ad libitum. Concentrate and DRDP were two, equal-sized allocations offered at 07:00 and 16:00. All animals were kept in separate pens, and free choices were available for water and mineral blocks. The experiment consisted of four periods consisting of 21 days each. During the first 14 days, all cattle were provided their respective feeds in their houses, whereas the cattle were moved to metabolism crates for feces and urine collection during the last seven days. The animals were individually recorded for daily feed intake. All cattle were adapted to the crates for two days before the start of the experiment. Feces and urine were collected during the last five days to determine the results of the feed utilization and nitrogen balances.

### 2.3. Sampling and Laboratory Analysis

Total fresh fecal samples were individually weighed daily and sampled at 5% to determine daily DM. About 100 g of feces was collected and stored in a refrigerator. This was mixed together from each animal on the last day of each period for the detection of chemical composition. Rice straw, DRDP concentrate, and feces were collected during the last seven days for analyzation of DM, organic matter (OM), ether extract (EE), crude protein (CP), ash, and acid detergent fiber (ADF) by the Association of Official Analytical Chemists (AOAC) method [10]. The neutral detergent fiber (NDF) in samples was estimated according to Van Soest et al. [11] with the addition of alpha-amylase, but without sodium sulfite, and the results are expressed with residual ash. The DM and chemical composition of the four experimental diets were offered as per the laboratory analysis outlined. Metabolizable energy (ME) was evaluated following the equation of Robinson et al. [12], whereas in vitro organic matter digestibility (ivOMD) values were obtained from our previous in vitro study with mean values of 530 g/kg DM. Urine was sampled in 10-L vessels containing 10% H_2_SO_4_ to ensure that the pH was decreased below 3.0 in order to prevent microbe destruction. Urine were pooled by cattle at the end of each period for nitrogen analysis [10] and calculation for and nitrogen balances.

Ten mL blood samples were taken from the jugular vein 0 and 4 h after feeding to determine the blood urea-nitrogen (BUN) (L typeWako UN, Tokyo, Japan). The blood samples were centrifuged at 500× *g* for 10 min at 4 °C and kept at −20 °C until analyzed. After blood sampling, 100 mL of ruminal fluid was sampled at approximately 0 h and 4 h after feeding through the mouth from the middle part of the rumen by a stomach tube connected to a vacuum pump. The ruminal fluid was immediately measured for pH and temperature (Hanna Instruments’ HI 8424 microcomputer, Kallang Way, Singapore). The ruminal fluid samples were filtered through four layers of cheesecloth. The ruminal fluid was separated into two parts: in the first part, 45 mL of ruminal fluid was mixed with 5 mL of 1 M H_2_SO_4_, centrifuged at 16,000× *g* for 15 min, and used for NH_3_–N estimation using the Kjeltech Auto 1030 Analyzer (Tecator, Höganäs, Sweden), and the volatile fatty acids (VFAs) were measured using high pressure liquid chromatography (HPLC) (Gilson Inc., Middleton, WI, USA). The last 1 mL proportion of ruminal fluid was mixed with a 9 mL solution (10% formalin solution in sterilized 0.9% saline) and used for the analysis of total fungal zoospores and protozoal counts using a hemocytometer [13] (Bulldog Bio, Inc., Portsmouth, NH, USA).

### 2.4. Statistically Determination

All data were subjected to analysis of variance according to a 4 × 4 Latin square design using General Linear Model (GLM) procedure of SAS (Version 6.0; SAS Inst. Inc. Cary, NC, USA). The results are presented as the mean values and standard error of the means. Means were compared using Duncan’s multiple range test. Differences among the means with *p* < 0.05 were accepted as representing statistically significant differences.

## 3. Results and Discussion

### 3.1. Feed Utilization

The influence of different levels of DRDP on the utilization of feed in Wagyu-crossbred cattle is displayed in Table 2. Roughage intake, total intake, and estimated energy intake varied significantly among the DRDP levels (*p* < 0.05), and superb intake occurred with the addition of 150 g/d DRDP. The higher feed intakes of 150 g/d DRDP compared with the no DRDP fed group could be attributed to the pelleting diet improving palatability and increasing feed intakes. Total apparent N excretion did not differ among the DRDP levels (*p* > 0.05), whereas N intake, apparent N absorption, and apparent N retention were significantly improved when compared to the no DRDP group (*p* < 0.05). The improvement of N utilization with DRDP indicates the positive practical effect of DRDP with high estimated N intake diets compared to the control diet (no DRDP) with N deficiency. The current results could provide a novel addition to the work reported by Seankamsorn et al. [5], who did not determine the influence of DRDP on N utilization efficiency. The apparent digestibility of NDF and ADF were similar (*p* > 0.05), despite the different DRDP levels. This might be due to the population of cellulolytic bacteria not being enhanced when DRDP was added; however, the current work did not study the cellulolytic bacteria. Thus, further study on cellulolytic bacteria diversity as affected by DRDP is needed. Apparent CP digestibility enhanced (*p* < 0.05) with the enhancing doses of DRDP supplementation. Inclusion of DRDP at 150 g/d produced the highest apparent digestibility of CP compared to the group that was not fed DRDP.

Tan et al. [8] found similar reports and demonstrated that Thai cattle had higher feed consumption with pellet supplementation than without it. Greater feed intake may be related to higher feed digestion when cattle were fed DRDP feeds. The apparent digestibility of CP was increased by 7.2% with 150 g/d DRDP supplementation. There is a possibility that DRDP contained sufficient nutrients such as energy (molasses), nitrogen (urea), and minerals as well as essential substances from rumen digesta to support greater synthesis of microbial cells, thus resulting in enhanced microbial activity to break down feed [14]. Furthermore, it could be due to the occurrence of active fungal zoospore, (Table 3), which has frequently been accompanied by enhanced CP digestion. Wallace et al. [15] indicated that a strain of the ruminal fungus *Neocallimastix frontalis,* isolated from the rumen of a sheep, had great proteolytic activity, which would support apparent CP digestibility. Cherdthong and Wanapat [16] revealed that the inclusion of 8% DRD could improve in vitro true digestibility by 11.5%. A previous study indicated that the addition of DRD in the diets of rabbits [6] or beef cattle [4] could increase feed intake and apparent digestibility when compared with the group that was not fed DRD. However, the dietary CP of cattle could supply adequate rumen function and microbial activity.

### 3.2. Rumen Parameters and Blood Metabolites

The effects of DRDP supplementation on the ruminal pH, temperature, NH_3_–N, BUN, and population of ruminal microorganisms are displayed in Table 3. DRDP inclusion did not change (*p* > 0.05) ruminal temperature and ruminal pH. However, the contents of rumen NH_3_–N and BUN were significantly different when compared to the no-supplement group (*p* < 0.05). Supplementation of DRDP did not alter the population of protozoa, whereas the addition of 150 g/d DRDP significantly increased the fungal zoospore. The mean values of rumen temperature and rumen pH were stable from 39.11 °C to 39.56 °C and 6.72 to 6.76, respectively, which were suitable ranges for the microbial activity to breakdown fibers and diet [4]. Furthermore, the rumen pH values in each treatment did not increase with the added DRDP, which could help avoid the confounding of pH with ruminal NH_3_–N effects. The ruminal NH_3_–N levels increased in response to added increments of DRDP. This may be related to the CP contents in DRDP that increased the rumen NH_3_–N. In the rumen, NH_3_ is incorporated as the byproduct of nitrogen into the carbon skeleton to yield glutamine and glutamate, which are essential nitrogen donors in nitrogenous compounds that metabolize in microbial cells. Therefore, increasing the doses of NH_3_–N with the increased doses of DRDP inclusion may also enhance microbial cell production. The averages of the ruminal NH_3_–N and BUN concentration for 150 g/d DRDP supplementation increased to 3.9 mg/dL and 2.4 mg/dL, respectively, compared to the group that was not fed DRDP. Nevertheless, the current findings agree with those of Leng [3], who suggested that the doses of ruminal NH_3_–N contents should be greater than 10 mg/dL for tropical zones to optimize the apparent digestibility.

The relationship between ruminal microbes can range from synergism to antagonism and depends on the group of microorganisms and related species and the type of substrate used [17]. There is now evidence to demonstrate the appearance of predacious and metabolic interference between fungal zoospores and protozoa in the rumen [18,19]. Fungal zoospores are of a similar size to bacteria that are digested by protozoa, thus it might be anticipated that protozoa would engulf fungal zoospores in the rumen. Protozoa also directly competes with fungi through enzyme chitinolytic destruction and predation [19]. In this study, the influence of DRDP level supplementation significantly altered the population of protozoa in the rumen of cattle (*p* < 0.05), which was reduced by 29.58% at 4 h post feeding when DRDP was supplemented at 150 g when compared to the control group. The lowering protozoal counts in ruminal fluid could be due to the highest NH_3_–N concentration, which has been reported by Kanjanapruthipong and Leng [20]. Protozoa did not utilize urea as a N source and amino acids are needed for protozoal synthesis [21]. This might potentially decrease protozoa synthesis in the rumen [20]. Moreover, a high concentration of NH_3_–N in the rumen may have a negative feedback mechanism on NH_3_–N diffusion from protozoa to inhibit protozoal growth. Leng [22] indicated that a sudden depression in protozoal numbers occurred in sheep given legumes with a high urea diet, thus resulting in improved feed utilization efficiency and N retention. Therefore, protozoa presence in the rumen may not be useful in ruminant feed with low quality diets because their enzymes’ beneficial effects might be counterbalanced by the sensitivity of enzyme anaerobic fungi to N shortage in the rumen [17,23]. Thus, it seems that a low number of protozoa enhanced apparent N absorption and apparent N retention as demonstrated in Table 2.

Fungal zoospores have been indicated to increase in the rumen when containing a low protozoal count. Romulo et al. [24] reported an increase in the zoospores population found in sheep removed rumen protozoa. Similarly, the current results indicated that an increasing population of fungal zoospores at 38.80%, (4 h post feeding) with an increased DRDP level of 150 g/d, might be related to the reduction trend of protozoal population. In addition, it could be related to many factors in DRDP that improved microbial cell synthesis in the rumen of Thai native, Wagyu-crossbred, beef cattle. Two mechanisms for improving fungal effects on consuming diet and improving apparent digestibility are feasible: (a) the supplement of a feed that is particular to the nutrition and expansion of the indigenous ruminal zoospores, and (b) the inoculation of high proficiency strains into the rumen [25]. Okpanachi et al. [6] noted that that the DRD contained microbial cells, essential amino acids, minerals, and non-negative physiological factors, which may beneficially affect ruminal fungal–cell synthesis. Ruminal zoospores are significant to cattle consuming diets of low-quality rice straw because they enhance the feed intake. The relationship is quite probably a result of fungi adhering to the lignified feed tissues, combined with the resultant softening of these tough-feed parts [25,26]. Thus, great potential exists for the improvement of fungi counts and activity in the rumen, which would enhance the use of low-quality rice straw among animals for enhanced yields. Moreover, enhancing the ruminal fungal zoospores with 150 g/d DRDP led to enhanced apparent CP digestibility. An amount of ruminal fungal zoospore generated extracellular proteinases in cultures. Obviously, ruminal fungal zoospore have an extensive range of proteolytic abilities, some of which might be approved to promote ruminal CP digestion [15].

### 3.3. Concentrations of Ruminal Volatile Fatty Acids

The concentration of rumen total VFAs and VFA profiles in cattle receiving various levels of DRDP are displayed in Table 4. VFA is synthesized in large amounts through rumen fermentation and is of significance in that it serves more than 70% of the energy supply for the ruminant. Thus, supplementary diets of ruminant feed with low quality roughage might be able to maintain suitable ruminal VFA concentration to meet energy requirements. Supplementation of DRDP in various doses did not alter (*p* > 0.05) the amount of total VFAs or the VFA profiles. In this study, the production of VFAs and their profiles were not changed among the diets. Total VFA concentrations in all DRDP levels ranged from 111 to 114 mmol/L and were close to those reported by Cherdthong et al. [4]. These results clearly show that the addition of DRDP does not adversely affect ruminal VFA production in Thai native, Wagyu-crossbred cattle. Similarly, Cherdthong et al. [4] noted that a concentrated mixture of DRD could be replaced with soybean meal in as much as 100% of cattle rations without negative effects on VFA concentrations. Furthermore, Seankamsorn et al. [5] demonstrated that the inclusion of DRDP levels significantly affected propionic acid concentrations, which were the highest among swamp buffalo when 150 g/d DRDP was added. These differences in the results might be due to the differences between the animal species. Swamp buffalo were more efficient than beef cattle in various functions such as nitrogen use and roughage digestion, and swamp buffalo had greater rumen NH_3_–N contents than the beef cattle, which led to increasing ruminal fermentation as well as increased ruminal VFA and an increased consumption of diet [27]. Therefore, the variations between Thai native, Wagyu-crossbred cattle and swamp buffalo may support the need to clarify the differences in their gastrointestinal tract abilities due to the ruminal end-products available for absorption and utilization by ruminant animals [28].

## 4. Conclusions

Generated DRD is a waste and is an environmental contaminant in most slaughterhouses in Thailand. DRDP supplementation at 150 g/d could improve ruminal NH_3_–N concentration, digestion of crude protein, apparent N absorption, and N retention as well as increase fungal zoospores in Thai-native, Wagyu-crossbred cattle fed with rice straw based diets. Nevertheless, more research should be conducted to provide additional data for possible implementations.

## Figures and Tables

**Table 1 animals-10-00056-t001:** Ingredient and chemical composition of the concentrate, dried rumen digesta pellets (DRDP), and rice straw.

Item	Concentrate	DRDP	Rice Straw
Ingredients, kg dry matter (DM)			
Cassava chips	60.00	-	
Dried rumen digesta *	-	80.00	
Sunflower oil	-	4.00	
Cassava starch	-	0.50	
Rice bran	13.00	-	
Coconut meal	12.00	-	
Palm meal	11.50	-	
Urea	2.50	10.00	
Molasses	1.00	2.50	
Sulfur	1.00	1.00	
Mineral and vitamin mixture ^a^	1.00	1.00	
Salt	1.00	1.00	
Chemical composition			
Dry matter, %	84.50	94.20	92.10
	%DM		
Organic matter	91.80	94.70	78.70
Crude protein	14.00	40.77	2.80
Neutral detergent fiber	30.10	53.10	68.10
Acid detergent fiber	14.70	31.10	40.70
Metabolizable energy (ME) ^b^, MJ/kg DM	10.51	8.65	7.23

^a^ Minerals and vitamins (each kg contains): vitamin A (10,000,000 IU), vitamin E (70,000 IU), vitamin D (1,600,000 IU), Fe (50 g), Zn (40 g), Mn (40 g), Co (0.1 g), Cu (10 g), Se (0.1 g), I (0.5 g); ^b^ Metabolizable energy (ME) was calculated according to the equation of Robinson et al. (2004); * Dried rumen digesta contains 19.42% crude protein, 92.5% organic matter, 42.35% neutral detergent fiber, and 20.54% acid detergent fiber.

**Table 2 animals-10-00056-t002:** Effect of dried rumen digesta pellets (DRDP) on feed intakes and apparent digestibility in Thai native, Wagyu-crossbred beef cattle.

Item	DRDP Supplementation (g/d)				SEM	*p*-Value
	0	50	100	150		
Feed intake (expressed on a dry matter)						
Rice straw						
kg/d	1.79 ^a^	2.29 ^a,b^	2.38 ^a,b^	2.47 ^b^	0.21	0.03
g/kg BW^0.75^	44.18 ^a^	55.50 ^b^	56.58 ^b^	61.05 ^c^	3.79	0.04
Concentrate						
kg/d	0.82	0.79	0.79	0.81	0.91	0.81
g/kg BW^0.75^	17.79	17.62	17.64	17.74	3.22	0.83
DRDP						
kg/d	0.00	0.05	0.10	0.15	-	-
g/kg BW^0.75^	0.00	2.53	4.95	6.74	-	-
Total intake						
kg/d	2.61 ^a^	3.13 ^b^	3.27 ^b^	3.43 ^c^	1.08	0.01
g/kg BW^0.75^	61.97 ^a^	75.65 ^a,b^	79.17 ^b^	85.53 ^c^	0.22	0.01
Nutrient intake						
Nitrogen (N) intake, g/d	26.4 ^a^	31.2 ^a,b^	34.9 ^b^	39.0 ^c^	1.56	0.03
Total N excretion, g/d	18.1	19.4	19.1	19.0	0.28	0.12
Fecal N excretion, g/d	7.6	8.2	8.1	8.0	0.61	0.11
Urinary excretion, g/d	10.6	11.2	11.1	11.0	0.66	0.45
Apparent N absorption, g/d	18.8 ^a^	23.0 ^b^	26.8 ^b^	31.0 ^c^	1.41	0.04
Apparent N retention, g/d	8.3 ^a^	11.8 ^a^	15.8 ^b^	20.0 ^c^	1.02	0.03
Apparent digestibility						
Dry matter, %	63.5	64.3	62.0	63.9	2.52	0.93
	------------------%DM-------------					
Organic matter	66.1	64.6	66.5	66.7	0.74	0.30
Crude protein	60.4 ^a^	64.5 ^a,b^	65.5 ^b^	67.6 ^b^	1.01	0.01
Neutral detergent fiber	53.0	53.8	54.6	56.4	1.20	0.30
Acid detergent fiber	45.5	47.0	45.8	44.8	1.15	0.61

SEM, standard error of the mean. ^a,b,c^ Means in the same row with different superscripts differ (*p* < 0.05).

**Table 3 animals-10-00056-t003:** Effect of dried rumen digesta pellets (DRDP) on rumen fermentation in Thai native, Wagyu-crossbred beef cattle.

Item	DRDP Supplementation (g/d)				SEM	*p*-Value
	0	50	100	150		
Ruminal pH						
0 h post feeding	6.76	6.79	6.76	6.73	0.05	0.88
4 h post feeding	6.74	6.72	6.72	6.70	0.02	0.57
Mean	6.75	6.76	6.74	6.72	0.02	0.70
Ruminal temperature, °C						
0 h post feeding	38.93	38.94	39.25	38.99	0.19	0.63
4 h post feeding	39.50	39.28	39.87	39.83	0.20	0.23
Mean	39.22	39.11	39.56	39.41	0.15	0.25
Ammonia-N concentration, mg/dL						
0 h post feeding	9.5	11.3	13.1	13.2	0.89	0.09
4 h post feeding	11.8 ^a^	12.9 ^a^	14.7 ^b^	14.8 ^b^	0.24	0.05
Mean	10.6 ^a^	12.1 ^a.b^	13.9 ^a,b^	14.0 ^b^	0.25	0.02
Blood urea-N, mg/dL						
0 h post feeding	9.3	10.0	10.3	11.8	0.89	0.08
4 h post feeding	10.3	11.5	11.8	12.5	0.98	0.13
Mean	9.8 ^a^	10.8 ^a^	11.0 ^a^	12.1 ^b^	0.27	0.02
Ruminal microbes, cell/mL						
Protozoa, ×10^5^						
0 h post feeding	1.50	1.63	1.38	1.25	0.43	0.70
4 h post feeding	2.13 ^a^	1.75 ^a,b^	1.63 ^a,b^	1.50 ^b^	0.10	0.03
Mean	1.82	1.69	1.51	1.38	0.57	0.20
Fungal zoospore, ×10^4^						
0 h post feeding	1.00	1.00	1.25	1.13	0.19	0.75
4 h post feeding	1.53 ^a^	1.63 ^a^	1.98 ^a,b^	2.50 ^b^	0.07	0.02
Mean	1.25 ^a^	1.31 ^a^	1.61 ^a,b^	1.81 ^b^	0.04	0.05

SEM, standard error of the mean. ^a,b^ Means in the same row with different superscripts differ (*p* < 0.05).

**Table 4 animals-10-00056-t004:** Concentrations of total volatile fatty acids (TVFAs) and VFA profiles of Thai native, Wagyu-crossbred beef cattle fed with various levels of dried rumen digesta pellets (DRDP).

Item	DRDP Supplementation (g/d)				SEM	*p*-Value
	0	50	100	150		
Total VFA, mmol/L						
0 h post feeding	111	112	109	114	3.96	0.79
4 h post feeding	113	115	111	115	2.01	0.46
Mean	112	114	110	115	2.06	0.42
Acetic acid, mol/100 mol						
0 h post feeding	61.81	60.54	62.04	61.82	2.17	0.96
4 h post feeding	63.70	62.25	66.98	63.57	2.57	0.63
Mean	62.75	61.40	64.51	62.69	1.85	0.71
Propionic acid, mol/100 mol						
0 h post feeding	21.48	23.51	22.40	21.28	2.17	0.88
4 h post feeding	24.36	23.58	25.37	25.13	2.25	0.94
Mean	22.92	23.54	23.89	23.21	2.01	0.99
Butyric acid, mol/100 mol						
0 h post feeding	12.48	14.96	12.06	11.90	2.01	0.69
4 h post feeding	14.11	17.92	15.66	13.81	2.89	0.74
Mean	13.29	16.44	13.86	12.86	2.30	0.70
Acetic acid:propionic acid ratio	2.90	2.69	2.38	2.85	0.33	0.70
Acetic plus butyric acid:propionic acid ratio	3.54	3.37	2.92	3.42	0.37	0.66

SEM, standard error of the mean.

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
