# Peer review of "Dried Rumen Digesta Pellet Can Enhance Nitrogen Utilization in Thai Native, Wagyu-Crossbred Cattle Fed Rice Straw Based Diets"

_animals, 2019, doi:10.3390/ani10010056_

Round 1

Reviewer 1 Report

Review of Animals-675516 "Dried rumen digesta pellet can enhance ruminal fungal zoospores and nitrogen utilization in Thai native, Wagyu-crossbred cattle fed rice straw based".

Major comments:

The main objective of this paper was to determine the effects of dried rumen digesta pellets on efficiency of feed utilization and rumen health in Thai native, Wagyu-crossbred cattle.

This manuscript suffers for substantial issues related to language. You need to ensure the logic of the full text. In your experimental design, the DRDP also contain Sunflower oil, Cassava starch, Urea, Molasses, Sulfur, Mineral and vitamin mixture except DRD, how to exclude the influence of other components? In results, you should indicate all the data of digestibility, absorptivity and deposition are the apparent index. You should explain your results through other studies, not just by saying that they are similar to previous studies. Your main conclusion is that increase the N utilization, so you need to emphatically discuss the mean of NH3-N and BUN (The article does not discuss changes in BUN). The conclusions are more a resume of the found results. This must be improved.

Minor comments:

1) Please choose a title that is accurate and concise. Focus on N utilization

2) Please standardized article citation format. Such as L38-39“Generated rumen digesta are neglected and compose serious natural contaminants in 38 slaughterhouses [4].” compared with L42-43“Cherdthong et al. [4] reported that DRD consists of crude 42 protein at 19.4% dry matter (DM).”

3) Please write full names and abbreviations where proper nouns first appear in the article.

4) The N absorption, N retention appearing in the text are all apparent N absorption and apparent N retention. Please correct.

5) L90: Please adjust the word order of “All animals were allowed time during the first 2 days of each period to adjust to their environments (pen and feed).”

6) L91: Please add: record feed intake per cow.

7) L132-133: How to get these results?

8) L139-143: In your experiment, the pelleted feeds just be used for DRD, not similar with other studies. So in your experiment, the effect of pelleted feeds on digestibility was not the main reason

9) L167-168: the contents compared to the concentration?

10) L170: What is the unit of the fungal zoospore?

11) L190-191: What is the mean of “indigested”?

12) L199: What is the mean of “there is a great potential for bacteria or fungi to occur and complete effectively for fermentable energy sources with the protozoa”? You need to redescribe why the lowering protozoal counts in ruminal fluid could be due to the highest NH3-N concentration.

13) L203-204: How to get these results of “As a result, the presence of protozoa has an adverse effect on the overall energetic proficiency of the rumen ecology”.So I suggested to redescribe the effects of protozoa on feed utilization efficiency and N retention.

14) Please check Table3 the Mean of Ammonia-N concentration of 0 g/d group.

Author Response

Response to Reviewer 1

Review of Animals-675516 "Dried rumen digesta pellet can enhance ruminal fungal zoospores and nitrogen utilization in Thai native, Wagyu-crossbred cattle fed rice straw based".

Major comments:

The main objective of this paper was to determine the effects of dried rumen digesta pellets on efficiency of feed utilization and rumen health in Thai native, Wagyu-crossbred cattle.

This manuscript suffers for substantial issues related to language. You need to ensure the logic of the full text.

Response: Thanks. Regarding to Reviewer 1 and other 2 Reviewer’s comments and suggestion, below we have tried our best to improve manuscript in order to prompt for publication. Please see details below.

In your experimental design, the DRDP also contain Sunflower oil, Cassava starch, Urea, Molasses, Sulfur, Mineral and vitamin mixture except DRD, how to exclude the influence of other components?

Response: Thank you very much for the valuable comment on DRDP ingredients. DRDP was consisted various ingredients due to we would like to improve more quality of the DRD such as high protein, energy and minerals. Thus, DRDP may be used as alternative feed supplement to ruminant. We agreed with you that these additional factor from each ingredient in DRDP may affected some parameters which we could not excluded the influence of other components. However, in section of result and discussion such as L117-119, L225-229, we tried to explained that the influence of other ingredients (urea, molasses, mineral et.) may also affect results. Thus, in our opinion present discussion might be made the reader clearer more.

In results, you should indicate all the data of digestibility, absorptivity and deposition are the apparent index. You should explain your results through other studies, not just by saying that they are similar to previous studies.

Response: Thanks so much and we have indicated “apparent” for digestibility, absorptivity and deposition. Furthermore, we have tried our best to provide explanation results where they needed to explain and also removed some explanation wherever not useful. Please see in manuscript.

Your main conclusion is that increase the N utilization, so you need to emphatically discuss the mean of NH3-N and BUN (The article does not discuss changes in BUN). The conclusions are more a resume of the found results. This must be improved.

Response: Now, we have modified section of conclusion and changed “N utilization” to “apparent N absorption and N retention” and the new version is “…DRDP supplementation at 150 g/d could improve ruminal NH3-N concentration, digestion of crude protein, apparent N absorption and N retention, as well as and increase fungal zoospores in Thai-native, Wagyu-crossbred cattle fed with rice straw based diets.” Please see in section of conclusion.

Minor comments:

1) Please choose a title that is accurate and concise. Focus on N utilization

Response: Thanks for the suggestion and now we have focused only on N utilization andchanged from “Dried rumen digesta pellet can enhance ruminal fungal zoospores and nitrogen utilization in Thai native, Wagyu-crossbred cattle fed rice straw based diets” to “Dried rumen digesta pellet can enhance nitrogen utilization in Thai native, Wagyu-crossbred cattle fed rice straw based diets” Please see in Title.

2) Please standardized article citation format. Such as L38-39“Generated rumen digesta are neglected and compose serious natural contaminants in 38 slaughterhouses [4].” compared with L42-43“Cherdthong et al. [4] reported that DRD consists of crude 42 protein at 19.4% dry matter (DM).”

Response: We confirmed that above citation were formatted regarding to Animals journal style. We have already checked with Guide for author.

3) Please write full names and abbreviations where proper nouns first appear in the article.

Response: Thanks. We have already provided full names and abbreviations where proper nouns first appear in the manuscript and please see in section of “Simple Summary:”, “Abstract”, etc. Please see throughout manuscript.

4) The N absorption, N retention appearing in the text are all apparent N absorption and apparent N retention. Please correct.

Response: We have already corrected regarding to the suggestion. Please see in section of “Abstract, Results and Discussion etc.

5) L90: Please adjust the word order of “

All animals were allowed time during the first 2 days of each period to adjust to their environments (pen and feed).”

Response: We have modified to “All cattle were adapted to the crates for two days before the start of the experiment.” Please see throughout manuscript.

6) L91: Please add: record feed intake per cow.

Response: We have added as “The animals were individual recorded daily feed intake.” Please see in manuscript.

7) L132-133: How to get these results?

Response: Thanks for comment. In order to make it not confuse, now we have removed this sentence from the section. Please see in section of “3.1 Feed utilization”

8) L139-143: In your experiment, the pelleted feeds just be used for DRD, not similar with other studies. So in your experiment, the effect of pelleted feeds on digestibility was not the main reason

Response: Thanks and we have agreed with comment. In order to make it not confuse, now we have removed these sentence from the section. Please see in section of “3.1 Feed utilization”

9) L167-168: the contents compared to the concentration?

Response: We have modified to “However, the contents of rumen NH3-N and BUN were significantly different compared to the no-supplement group (P < 0.05).” Please see in manuscript.

10) L170: What is the unit of the fungal zoospore?

Response: The unit of the fungal zoospore is “cell/mL” and we have provided in the same line of “Ruminal microbes, cell/mL”. Please see in Table 3.

11) L190-191: What is the mean of “indigested”?

Response: Sorry, we have modified to “Fungal zoospores are of a similar size to bacteria that are digested by protozoa, thus it might be anticipated that protozoa would engulf fungal zoospores in the rumen.” Please see in manuscript.

12) L199: What is the mean of “there is a great potential for bacteria or fungi to occur and complete effectively for fermentable energy sources with the protozoa”? You need to redescribe why the lowering protozoal counts in ruminal fluid could be due to the highest NH3-N concentration.

Response: Thanks and we have agreed with comment. In order to make it not confuse, now we have removed this sentence from the section and modified as “The lowering protozoal counts in ruminal fluid could be due to the highest NH3-N concentration, which has been reported by Kanjanapruthipong and Leng [20]. Protozoa did not utilize urea as a N source and amino acids are needed for protozoal synthesis [21]. This might potentially decrease protozoa synthesis in the rumen [20].”. Please see in section of “3.2. Rumen Parameters and Blood Metabolites”

13) L203-204: How to get these results of “As a result, the presence of protozoa has an adverse effect on the overall energetic proficiency of the rumen ecology”.So I suggested to redescribe the effects of protozoa on feed utilization efficiency and N retention.

Response: Thanks and we have agreed with comment. In order to make it not confuse, now we have removed this sentence from the section and modified as “Leng [22] indicated that a sudden depression in protozoal numbers occur in sheep given legumes with a high urea diet, thus resulting in improved feed utilization efficiency and N retention. Therefore, protozoa presence in the rumen may not be useful in ruminant feed with low quality diets because their enzyme’s beneficial effects might be counterbalanced by the sensitivity of enzyme anaerobic fungi to N shortage in the rumen [17,23].” Please see in manuscript.

14) Please check Table3 the Mean of Ammonia-N concentration of 0 g/d group.

Response: Thanks for your great suggestion. The mean of ammonia-N concentration of 0 g/d group is correct, but at 0 h post feeding was modified from 10.66 to 9.48 mg/dL. Please see in Table 5, page 6.

Thank so much!

Reviewer 2 Report

The overall quality of this manuscript has grown considerably and I emphasize that the modifications have tremendously affected its readability. As previously stated, I strongly believe that the use of “Dried Rumen Digesta” not only represents a novelty in the area, but it is very important because it demonstrates that an waste product could (and should) be converted into human edible protein. Again, my conclusion is that the current manuscript presents sufficient originality and substance to be worthy of publication and the modifications made so far had greatly increased the quality of the manuscript.
Despite this, I still have some minor suggestions that I list below in an attempt to help the authors further improve their manuscript:
Line 11 – “feed residues feed”. The word feed appears twice, please delete the second one.
Line 11 – Dried rumen digesta “show”? Consider replacing the word show for have (if plural) or has (if singular).
Line 32 – I suggest you replace the expression “fatten-beef cattle” for: enterprises “lot feeding cattle”, or “feeding cattle”, or “finishing cattle”. Note: It is just a suggestion, the currently used terminology seems weird to me.
Line 38 – Replace “compose” for “represent”. Note: Compose means to create, but DRD are already a natural contaminant as is.
Line 68 –“additional small level of concentrate diet”. Delete the word diet.
Line 189 – metabolic “interfere”? Did you mean “interference”?
Line 191 – You are saying that fungal zoospore are of a similar size to bacteria that are “indigested” by protozoa, anticipating that protozoa would engulf fungal zoospores. Note: I believe you meant the opposite.

Author Response

Response to Reviewer 2

The overall quality of this manuscript has grown considerably and I emphasize that the modifications have tremendously affected its readability. As previously stated, I strongly believe that the use of “Dried Rumen Digesta” not only represents a novelty in the area, but it is very important because it demonstrates that an waste product could (and should) be converted into human edible protein. Again, my conclusion is that the current manuscript presents sufficient originality and substance to be worthy of publication and the modifications made so far had greatly increased the quality of the manuscript.

Response: We appreciate thanks to the Reviewers 2 to see our effort revised this manuscript and suggested that our work is the novel and present the valuable information to the readers. Furthermore, in order to improve more our manuscript we have more revise accordingly to the comment made by you and also from the other two Reviewers. Thus, current revise version we expected that it should be ready for publication. Thanks!

Despite this, I still have some minor suggestions that I list below in an attempt to help the authors further improve their manuscript:

Response: Thanks, please see below which we have tried our best to revise accordingly to your comments.

Line 11 – “feed residues feed”. The word feed appears twice, please delete the second one.

Response: We have deleted. Please see in manuscript.

Line 11 – Dried rumen digesta “show”? Consider replacing the word show for have (if plural) or has (if singular).

Response: We replaced by “have”. Please see in L11.

Line 32 – I suggest you replace the expression “fatten-beef cattle” for: enterprises “lot feeding cattle”, or “feeding cattle”, or “finishing cattle”. Note: It is just a suggestion, the currently used terminology seems weird to me.

Response: We have changed this sentence as “A number of successful enterprises finishing cattle have started to supply high-quality beef.” Please see in manuscript.

Line 38 – Replace “compose” for “represent”. Note: Compose means to create, but DRD are already a natural contaminant as is.

Response: We have replaced. Please see in manuscript.

Line 68 –“additional small level of concentrate diet”. Delete the word diet.

Response: We have deleted. Please see in manuscript.

Line 189 – metabolic “interfere”? Did you mean “interference”?

Response: We have already revised as “interference”. Please see in manuscript.

Line 191 – You are saying that fungal zoospore are of a similar size to bacteria that are “indigested” by protozoa, anticipating that protozoa would engulf fungal zoospores. Note: I believe you meant the opposite.

Response: We have modified to “Fungal zoospores are of a similar size to bacteria that are digested by protozoa, thus it might be anticipated that protozoa would engulf fungal zoospores in the rumen.” Please see in manuscript.

Thank so much!

Reviewer 3 Report

Authors have made a great effort to improve their manuscript. Some details need attention before considering this manuscript for publication:

Please add the word diets at the end of the title and whenever appropriate in the main text and/or tables Remove rumen ecology and replace it with rumen microorganisms. Ecology is a concept and as far as I know you data is not showing anything about rumen ecology i.e., alpha and beta diversities, differences in micro communities. You have measured total protozoa and fungal zoospores, so, it will be too ambitious to say rumen ecology.

Author Response

Response to Reviewer 3

Authors have made a great effort to improve their manuscript. Some details need attention before considering this manuscript for publication:

Response: Thanks so much that you can see our effort in order to improve manuscript and ready to publication. Below, we have response to all suggestion by Reviewer already. Please see throughout manuscript.

Please add the word diets at the end of the title and whenever appropriate in the main text and/or tables

Response: Thanks, and we have added wherever appropriated. Please see throughout manuscript.

Remove rumen ecology and replace it with rumen microorganisms. Ecology is a concept and as far as I know you data is not showing anything about rumen ecology i.e., alpha and beta diversities, differences in micro communities. You have measured total protozoa and fungal zoospores, so, it will be too ambitious to say rumen ecology.

Response: We have replaced rumen ecology with rumen microorganisms already. Please see throughout manuscript.

Thank so much!

This manuscript is a resubmission of an earlier submission. The following is a list of the peer review reports and author responses from that submission.

Round 1

Reviewer 1 Report

Review of Animals-644309 " Dried rumen digesta pellet can enhance ruminal characteristics and diet use in Thai native, Wagyu-crossbred cattle". General comments: The main objective of this paper was to determine the effects of dried rumen digesta pellets on efficiency of feed utilization and rumen health in Thai native, Wagyu-crossbred cattle. First of all, from the perspective of experimental innovation, authors just do it on another type of animal compared with other researchers, authors were not really able to use the introduction to justify the significance of the study. In the experimental design, authors use the 4×4 Latin square experimental design, 21 days every periods was not properly justified. About results, authors did not calculate the nitrogen loss in the urine and could not judge the nitrogen retention, so this part of the data is unreliable. The increase in the concentration of NH3-N in the rumen is not necessarily beneficial to ruminants, and the optimal concentration should be determined in combination with nitrogen emissions from the urine. Therefore, nitrogen in urine is an important indicator. In addition, authors need to use literature support to explain the role of Protozoa and Fungal zoospore in the rumen. Most importantly, authors need to check the accuracy of the data and provide a calculation method(e.g. Table 3, the accuracy of BUN Mean.)The concentration of VFA should also be discussed, not just the VFA profiles. In short, the data of this manuscript is insufficient to support the author's purpose and conclusion. The English must also be improved. I would recommend a native English speaker to review the paper in detail. So I don't recommend this work to be published in the journal. Minor comments: L2-3: What does “ruminal characteristics” include? Feed utilization instead of diet use L11: “mainly found in ruminant animals” Delete. L22: Supplementation instead of Inclusion L26: DRDP instead of DRD in pellet form, Check other instances L35: abattoir need standardization L42: without instead of with no L52-59: What is the relationship between crossbreeding and this article? L73-77: Whether to use the same batch of DRDP throughout the experiment? L89: ad libitum need standardization L91: All cattles L101-103: Results of Phosphorus contents and Calcium contentswere either not mentioned or discussed/interpreted. L108-109: Need to explain whether to sample through the mouth. L125: doses is not suitbale L130: How to reflect the N balance? L143: How did you calculated N intake, N absorption,N excretion, N retention and Metabolizable energy? Please study the definition of metabolic energy carefully. L149-154: These results cannot be derived from Table 3. Please use a lot of literature to explain the phenomenon you found. L176-177: “In addition, the increased CP digestion 176 associated with 150 g/d DRDP may also affect the rumen NH3-N concentration” Same as the previous sentence. L180: Please carefully check the accuracy of the data in Table 3! L183-184: This sentence is not clear. L201-202: What is the mean of “the CP was participatory with 201 diet particles”? L212: Due to the experimental results of Seankamsorn et al. So you need discuss the concentration of VFA, not just the VFA profiles. L221: Please note the format of Table 4. L227: Can it improve rumen fermentation?

Reviewer 2 Report

The work is relevant to the field of Animal Nutrition and warrants consideration for publication. The use of “Dried Rumen Digesta” not only represents a novelty in the area, but also has an appeal considering that a waste product would be converted into human edible protein i.e. beef. My conclusion is that the current manuscript presents sufficient originality and substance to be worthy of publication. Despite this, I recommend that modifications are made to improve the quality of the manuscript, with a particular emphasis on the use of the English language. Examples: Line 32. … growing consumption “in” ruminant meat (“of” ruminant meat). Line 90. All “beef” were kept in “segregate” houses (All “animals” were kept in “segregated” houses à meaning houses set a part). Overall, the appropriateness of the experimental design and analyses appear to be sound, but some aspects need further clarification.

Below I list some suggestions and raise questions in an attempt to help the authors improve the overall quality of their manuscript:

The “Introduction” section starts well, but at around Line 52 it loses focus from the main topic, that is the nutritional value of DRD. The authors start talking about crossbreeding and the potential effects on marbling. That is all fine, but it is not the topic of the current manuscript. There was only one type of animal utilized, so no comparisons between different breeds and there were no analysis of meat parameters or longer chain fatty acids in the rumen fluid. Therefore, in my opinion the authors should focus exclusively on what they analysed. Also, I’ve noticed that in the Introduction (Line 40) the authors make reference to a work in which the CP content of DRD was 19.4% DM. The DRD of the current manuscript had twice this value, i.e. 40.77%. Perhaps it would be worth adding an item on the “Results and Discussion” session covering the variability between DRD found in the literature. I am confident that it would vary considerably depending on the original feed of the cattle that had been slaughtered to provide the rumen digesta.

Questions:

Why the use of the term “total N excretion”? Did you measure N lost in urine as well? If so, I could not find it in the manuscript.

In addition, in Line 133 you indicate that DRDP resulted in greater microbial protein synthesis. How did you conclude that? You used the rumen fluid samples to evaluate microbial populations, but I could not find a reference to your methods to quantify the microbial population.

Reviewer 3 Report

The manuscript animals-644309 was carried out to determine the effects of a diet of dried rumen digesta pellets (DRDP) on feed digestion, ruminal ecology, and ruminal microbes in Thai native, Wagyu-crossbred cattle. The manuscript English style MUST be checked. The data is of great value, however, due to the many English mistakes, it is difficult to follow up the main messages.

INTRODUCTION: Needs to be rewritten. Authors need to explain why and how did they come up with the objective. Additionally, a hypothesis needs to be indicated. It is not clear the link between feed digestion, ruminal ecology, and ruminal microbes. Also, conceptually, rumen ecology and rumen microbiome could be something different. Authors need to explain what has been done in those topics previously, and the relevance of doing research in those topics.

RESULTS AND DISCUSSION: A better flow and coherence is needed. Authors need to link how nutrients created an environment (i.e., substrates) where microorganisms grew and had their metabolism. Then they need to explain how protozoa or fungal zoospores interacted or antagonized. Please be aware that if you talk about ecology, then the level and the scope of your discussion may be different.

Lines 32-33 rewrite and add reference

Lines 33-35 check English style

Line 37 replace terrible for another word

Line 118 did you use a post-hoc test? Please explain

Lines 225-229 please give numbers or ranges when you say that DRDP improves ruminal fermentation, the

intake of low-quality roughage, and fungal zoospores